# Comparing the Election Systems for Overseas Constituency Representatives in Multiple Countries

**Shuji Yamauchi * and Takashi Sekiyama** 

Graduate School of Advanced Integrated Studies in Human Survivability, Kyoto University,
Kyoto 606-8306, Japan; sekiyama.takashi.2e@kyoto-u.ac.jp
* Correspondence: yamauchi.shuji.47m@st.kyoto-u.ac.jp

**Abstract:** Although electoral systems are a traditional focus in political science, limited research exists on the characteristics of overseas constituency representation. This study aims to quantitatively elucidate these characteristics through a comparative analysis of the election systems in eight countries. This study analyzes overseas constituency representative systems while focusing on key factors such as the number of eligible voters, seats, voter turnout, and representativeness (value of a single vote). Voter turnout in overseas districts varies significantly among these countries. Notably, Croatia and Romania exhibit exceptionally high voter turnouts in overseas districts. Common characteristics in high-turnout countries include a higher representativeness in overseas districts than the home country and a small proportion of overseas voters in the total electorate. This dynamic incentivizes overseas voters to participate in elections to reflect their minority opinions in national politics. Furthermore, it potentially leads to a higher voter turnout in overseas districts than in the home country.

**Keywords:** overseas constituencies; diaspora; election systems

## 1. Introduction

The retention of the rights and obligations of nationals residing abroad, as well as the preservation of their political and cultural ties with their home country, are becoming an increasingly important social issue for many countries due to the rising number of such individuals. According to the *World Migration Report* of the International Organization for Migration (IOM), there has been an overall increase in the number of international migrants over the past 50 years. In 2020, the total population living outside their country of origin was estimated to be 281 million—an increase of 128 million since 1990, and more than three times the 1970 estimate (IOM 2022) (Mcauliffe and Triandfyllidou 2022). The practical challenges that democratic nations face in the current era of increased human mobility include determining whether the political rights of these expatriates can be guaranteed, if their opinions can be reflected in national policies, and whether citizens living abroad can maintain an identity connection with their home countries.

In this context, there are some countries where non-residents are permitted to participate in the electoral contests. According to Wellman et al. (2023), a total of 141 countries have legally enfranchised non-resident citizens. However, only a limited number of countries allow nationals living abroad to directly elect representatives who then participate in the home country's parliament. These representatives are expected to invigorate discussions both domestically and internationally by maintaining political ties between the home country and its citizens abroad. The authors' research confirms that eight countries—France, Italy, Croatia, Romania, Ecuador, Peru, Cape Verde, and Tunisia—have implemented systems for electing representatives from overseas constituencies.

This study conducted a comparative analysis of the systems for electing representatives from overseas constituencies in the eight countries to quantitatively elucidate their characteristics. Although the analysis of electoral systems has been a traditional research

topic in political science, and the literature on extraterritorial voting system has been growing, there is a scarcity of research on systems for electing representatives from overseas constituencies. Existing research has rarely analyzed the characteristics and challenges of overseas electoral district systems in various countries. Particularly, as discussed later, Croatia and Romania demonstrate remarkably high voter turnouts in their overseas electoral districts. Elections are an indispensable process in democratic nations, and the voter turnout level is a crucial factor influencing a country's political process. A high voter turnout in overseas constituencies likely indicates that expatriate voters possess a keen interest and strong willingness to actively participate in their home country's political processes. The question arises as to why there is such a significant variation in voter turnout rates across countries, both within overseas constituencies and between domestic and international turnout rates.

Examining the factors influencing voter turnout at the national, regional, and local levels has emerged as a prominent research theme in political science. Previous studies have consistently shown that countries with compulsory voting systems, where citizens are legally mandated to vote, tend to exhibit higher turnout rates (Henderson and McEwen 2015; Franklin and Hobolt 2011; Rose and Borz 2013). Additionally, factors such as the significance of elections, where a single vote can impact legislative or parliamentary seats, or the occurrence of multiple elections on the same day (Blais and Dobrzynska 1998; Stockemer and Scruggs 2012), along with characteristics such as the homogeneity among citizens, close citizen–representative relationships, or relatively shorter geographical distances between citizens and the government (Kostadinova and Power 2007) have been identified as trends leading to a higher voter turnout. However, the exceptionally high turnout rates observed in countries that do not have compulsory voting such as Croatia and Romania raise intriguing questions. However, the impacts of other factors such as electoral system type, number of political parties, economic development, and income disparity on voter turnout remain inconclusive (Stockemer 2017).

Regarding turnout abroad, Hutcheson and Arrighi (2015) showed that it was influenced by a variety of institutional constraints restricting the scope of the policy (through residence and professional qualifications); limiting eligible voters' access to the ballot (through cumbersome registration procedures and voting methods); and reducing the electoral weight that is attributed to their votes (through distinct modes of representation). In this context, for example, a study on external voting by immigrants in Portugal, one of the EU countries with the highest emigration rates across the 15 main countries of residence, found that the method of voting and socioeconomic factors significantly influenced the voter turnout (Belchior et al. 2018). Burgess and Tyburski (2020) found that party mobilization increases extraterritorial voter turnout. According to their survey, non-resident citizens who maintain connections to their country of origin are more likely to vote in homeland elections. Peltoniemi et al. (2023) argue that individual-level conditions and social relations also play a central role in the turnout abroad. They found that non-resident citizens who maintained economic, social, and cultural connections to their country of origin were more likely to vote in homeland elections in the case of Finland.

However, the factors affecting the turnout in overseas constituencies remain unclear. Thus, this study aims to compare and analyze data on the number of voters, seat allocation (number of seats), voter turnout, and representativeness (value of a single vote) in overseas constituency representative systems across eight countries that have implemented systems for electing representatives from overseas constituencies. This study explores the factors influencing voter turnout rates in these overseas constituencies and the variations between the domestic and international turnout rates in these eight countries.

## 2. Materials and Methods

### 2.1. Countries Surveyed

The authors' research confirms that several European nations, including France, Italy, Croatia, and Romania, have adopted a system of electing representatives from overseas

constituencies, integrating the voices of their expatriate citizens into the political process. Furthermore, countries such as Ecuador and Peru in the Central and South American regions and Cabo Verde and Tunisia in the African region have implemented such systems. A recent study (Wellman et al. 2023) showed that other countries used or currently adopted a similar system. The system of electing representatives for overseas electoral districts has been confirmed to have been adopted in various countries since its first implementation in Portugal in 1976. However, there are cases where legislation was passed but the system was not actually put into effect, as seen in Angola and Panama (Collyer 2014). Furthermore, the system was abolished in the Cook Islands due to financial burdens (Hassall 2007), as well as in Morocco (Belguendouz 2004). Furthermore, although the system appears to still be in place in Portugal, its confirmation relies on limited and uncertain information, leading to the exclusion of this country from the scope of this study. Thus, this study focuses on the eight countries. Tables 1 and 2 summarize the system for electing representatives from overseas constituencies in these eight nations.

### 2.2. Data Sources

Data on the number of eligible voters, the number of seats (constituencies), and voter turnout in both overseas and domestic electoral districts for these eight countries were obtained from the Constituency-Level Elections Archive (CLEA), a database provided by the University of Michigan's Center for Political Studies. The CLEA serves as a repository of detailed electoral results at the constituency level for both upper- and lower-house elections worldwide. The data from the CLEA, publicly available as a comprehensive and reliable resource for national election results worldwide, are utilized by a diverse global audience for research, education, and policymaking purposes (CLEA 2022).

### 2.3. Data Processing

To elucidate the characteristics of the overseas electoral district representative system in each country, further calculations were performed based on the number of eligible voters, the number of seats (constituencies), and the voter turnout for both domestic and overseas electoral districts, obtained from the CLEA. The following six metrics were used to compare the domestic and overseas electoral districts in each country:

1. Representativeness (value of a single vote).
2. Proportion of overseas voters among eligible voters.
3. Number of overseas electoral district seats out of the total number of seats.
4. Differences in voter turnout between domestic and overseas districts.
5. Ratio of voters per representative in domestic versus overseas districts.
6. Average values of each metric across countries.

Table 3 summarizes all the indicators for each of the eight countries under study, encompassing the number of eligible voters, voter turnout, and representativeness, among others, for both domestic and overseas electoral districts.

Table 1. Overview of unicameral parliamentary countries and domestic and overseas constituency representative election systems.

| Country | Election Name | Duration of Term (Years) | Composition (Numbers) | Election District | Election System | Number of Seats | Composition | Number of Votes | Voting Scheme |
|---|---|---|---|---|---|---|---|---|---|
| Ecuador | National Assembly Election | 4 | 137 | Provincial constituents | Open-List proportional representation | 116 | 24 | 1 | Candidate |
| | | | | National List | Open-List proportional representation | 15 | 1 | 1 | Candidate |
| | | | | Overseas constituency (Diaspora) | Direct election/plurality voting | 6 | 3 | 1 | Candidate |
| Peru | Upper House of the National Legislature—National Assembly Election | 5 | 130 | Provincial constituents | Open-List proportional representation | 128 | 26 | 1 | Candidate |
| | | | | Overseas donstituency (Diaspora) | Open-List proportional representation | 2 | 1 | 1 | Candidate |
| Tunisia | Assembly of Peoples representation election | 5 | 161 | Multi-member constituencies | Individual voting, plurality/Majority | 151 | 27 | 1 | Candidate |
| | | | | Overseas constituency | Individual voting, plurality/Majority | 10 | 6 | 1 | Candidate |
| Cabo Verde | Parliamentary Election | 4 | 72 | Multi-member constituencies | Closed-Party-list proportional representation. | 66 | 10 | 1 | Candidate |
| | | | | Overseas constituency (Diaspora) | Closed-Party-list proportional representation | 6 | 3 | 1 | Candidate |
| Croatia | Parliamentary Election | 4 | 151 | Multi-member territorial constituencies | Closed-Party-list proportional representation | 140 | 10 | 1 | Candidate |
| | | | | Overseas constituency | Closed-Party-list proportional representation | 3 | 1 | 1 | Candidate |
| | | | | Minority deputies | Closed-Party-list proportional representation | 8 | 1 | 1 | Candidate |

Source: created by authors using government materials from each country.

**Table 2.** Overview of bicameral parliamentary countries and domestic and overseas constituency representative election systems.

| Country | Election Name | Duration of Term (Years) | Composition (Numbers) | Election District | Election System | Number of Seats | Composition | Number of Votes | Voting Scheme |
|---|---|---|---|---|---|---|---|---|---|
| France | French Senate | 6 | 348 | Metropolitan France | Two-Round system, Indirect universal suffrage, Proportional representation | 326 | | 1(2) | Candidate |
| | | | | Overseas departments and territories | | 10 | 128 | 1(2) | Candidate |
| | | | | Overseas constituency (Diaspora) | Proportional representation | 12 | | 1(2) | Candidate |
| | Legislative Election | 5 | 577 | Metropolitan France | Two-Round system | 539 | 21 | 1(2) | Candidate |
| | | | | Overseas departments and territories | Two-Round system | 27 | 5 | 1(2) | Candidate |
| | | | | Overseas constituency (Diaspora) | Two-Round system | 11 | 4 | 1(2) | Candidate |
| Italy | Senate of the Republic | 5 | 206 | Italian Parliament | Closed-Party-list proportional representation and plurality | 196 | 20 | 1 | Party or candidate |
| | | | | Senators for life | Permanent/Appointed by the President of the republic | 6 | 1 | 1 | Candidate |
| | | | | Overseas constituency (Diaspora) | Closed-Party-list proportional representation and plurality | 4 | 4 | 1 | Party or candidate |
| | Chamber of Deputies | 5 | 400 | Single-member constituency | Plurality voting | 147 | | 1 | Candidate |
| | | | | Multi-member constituency | National proportional voting | 245 | 27 | 1 | Party |
| | | | | Overseas constituency (Diaspora) | Constitutional proportional constituency | 8 | 4 | 1 | Party |
| Romania | Senate of the Republic | 4 | 136 | Senate of Romania | Party-list proportional representation | 134 | 42 | 1 | Party |
| | | | | Overseas constituency (Diaspora) | Party-list proportional representation | 2 | 1 | 1 | Party |
| | Chamber of Deputies | 4 | 330 | Deputies | Direct popular vote using party-list proportional representation | 308 | | 1 | Candidate |
| | | | | Minority deputies | Appointed | 18 | 42 | 1 | Candidate |
| | | | | Overseas constituency (Diaspora) | Direct popular vote using party-list proportional representation | 4 | 4 | 1 | Candidate |

Source: created by authors using government materials from each country.

**Table 3.** Indicators for countries that have adapted their overseas constituency representative system (summary).

| Index | Countries | | | | | | | | | Average |
|---|---|---|---|---|---|---|---|---|---|---|
| | Italy | Croatia | France (R1) | France (R2) | Romania | Cape Verde | Ecuador | Peru | Tunisia | |
| Number of voters in domestic constituencies | 46,505,350 | 3,643,765 | 46,529,339 | 44,570,109 | 18,285,773 | 302,942 | 11,384,216 | 22,901,954 | 8,273,068 | 22,488,502 |
| Number of seats in domestic electoral districts (number of seats) | 618 | 140 | 566 | 566 | 308 | 66 | 131 | 130 | 206 | 303 |
| Voter turnout in domestic constituencies | 72.94% | 46.02% | 49.44% | 43.5% | 39.44% | 70.29% | 81.79% | 81.88% | 41.49% | 1 |
| Number of voters per member of domestic constituency | 75,251.38 | 26,026.89 | 82,207.31 | 78,745.78 | 59,369.39 | 4590.03 | 86,902.41 | 176,168.88 | 40,610.520 | 69,936 |
| Number of voters in overseas constituencies | 4,230,854 | 58,159 | 1,264,845 | 1,265,237 | 117,089 | 44,680 | 285,753 | ※ | 721,596 | 998,527 |
| Number of seats in overseas electoral districts (number of seats) | 12 | 11 | 11 | 11 | 4 | 6 | 6 | 1 | 18 | 9 |
| Voter turnout in overseas constituencies | 29.84% | 99.92% | 19.11% | 16.44% | 94.46% | 36.69% | 46.60% | ※ | 15.52% | 44.82% |
| Number of voters per member in overseas constituencies | 352,571.17 | 5287.18 | 114,985.91 | 115,021.55 | 29,272.25 | 7446.67 | 47,625.50 | - | 40,088.67 | 89,037 |
| Proportion of overseas voters in total electorate | 8.34% | 1.57% | 2.65% | 2.76% | 0.64% | 12.85% | 2.45% | - | 8.02% | 4.91% |
| Proportion of overseas constituency seats in total number of seats | 1.90% | 7.28% | 1.91% | 1.91% | 1.28% | 8.33% | 4.38% | | 8.04% | 2.85% |
| Difference in voter turnout between domestic and overseas constituencies | 43.11% | −53.90% | 30.33% | 27.04% | −55.02% | 33.60% | 35.19% | - | 25.97% | 10.79% |
| Comparison of number of voters per representative between domestic and overseas constituencies. | 0.213 | 4.923 | 0.715 | 0.685 | 2.028 | 0.616 | 1.825 | - | 1.002 | 1.50 |
| Extraction year | 2018 | 2020 | 2017 | 2017 | 2016 | 2016 | 2013 | 2020 | 2014 | |

Source: created by the authors using CLEA data. Note: in France, due to the adoption of a two-round voting system, "R1" represents the first round of voting, while "R2" denotes the second round. ※ Note: Peruvian data are included in the resident count of the capital, Lima, and cannot be extracted separately.

## 3. Results

### 3.1. Comparison of Voter Turnouts in Overseas Constituencies

Voter turnouts in overseas constituencies vary significantly by country, with a mean turnout rate of 44.82%, ranging from 99.92% in Croatia to 15.52% in Tunisia. Table 4 illustrates the distribution of countries into groups with high, low, and moderate voter turnouts.

**Table 4.** Grouping by overseas constituency voter turnout.

| Criterion | Countries Targeted |
|---|---|
| Countries with higher-than-average voter turnout. | Croatia (99.92%), Romania (94.46%) |
| Countries with average voter turnout. | Ecuador (46.60%), Cape Verde (36.69%) |
| Countries with lower-than-average voter turnout. | Italy (29.84%), Tunisia (15.52%) France (1R: 19.11%, 2R: 16.44%) |
| Average turnout. | 44.82% |

Source: created by the authors using CLEA data. Note: Peruvian data are excluded due to unavailability.

### 3.2. Comparison of Voter Turnout Differences between Domestic and Overseas Constituencies

An examination of the differences in voter turnouts between domestic and overseas constituencies in various countries has revealed significant disparities. As shown in Table 5, while in many countries, the voter turnout in domestic constituencies surpasses that in overseas constituencies, in countries such as Croatia and Romania, the situation is reversed. In these countries, overseas constituencies have voter turnouts of 99.92% and 94.46%, respectively, significantly exceeding those in domestic constituencies.

**Table 5.** Groupings based on differences in voter turnout between domestic and overseas constituencies.

| Criterion | Countries Targeted |
|---|---|
| Countries with higher voter turnout in domestic constituencies | Ecuador (81.79%), Italy (72.94%), Cape Verde (70.29%), France (1R: 49.44%, 2R: 43.5%), Tunisia (41.49%) |
| Countries with higher voter turnout in overseas constituencies | Croatia (99.92%), Romania (94.46%) |

Source: created by the authors using CLEA data. Note: Peruvian data are excluded due to unavailability.

### 3.3. Comparison of the Percentage of Overseas Voters of All Voters

Comparing the percentage of overseas voters to all voters in each country, the average value was approximately 4.91%. Naturally, large variations are found between countries (see Table 6). Cape Verde is the most prominent example, with approximately 12.85% of all voters being overseas voters. Italy and Tunisia also had high percentages of overseas voters (8.34% and 8.02%, respectively). By contrast, the percentages remained low in Croatia and Romania at 1.57% and 0.64%, respectively.

**Table 6.** Groupings based on the percentage of overseas voters of total electorate.

| Criterion | Countries Targeted |
|---|---|
| Countries with higher-than-average percentage. | Cape Verde (12.85%), Italy (8.34%), Tunisia (8.02%) |
| Countries with average percentage. | France (1R: 2.65%, 2R: 2.76%), Ecuador (2.45%). |
| Countries with lower-than-average percentage. | Croatia (1.57%), Romania (0.64%). |
| Average value. | 4.91% |

Source: created by the authors using CLEA data. Note: Peruvian data are excluded due to unavailability.

### 3.4. Comparison of the Percentage of Seats in Overseas Constituencies of All Seats

Table 7 compares the percentage of seats that are held in overseas constituencies to the total number of seats in each country, indicating an average value of approximately

2.85%, with significant variations between countries. Among them, Cape Verde (8.33%) and Tunisia (8.04%) exceeded 8%. Along with Croatia's 7.28%, their percentage of seats in overseas constituencies was significantly higher than the average. Ecuador also had a rate of 4.38%, which is above the average. However, Italy and France had rates slightly below 2%, close to the average, and Romania was the only country with a low percentage of overseas seats (1.28%).

**Table 7.** Groupings based on the percentage of seats in overseas constituencies of all seats.

| Criterion | Countries Targeted |
| --- | --- |
| Countries with higher-than-average percentage. | Cape Verde (8.33%), Tunisia (8.04%), Croatia (7.28%), Ecuador (4.38%). |
| Countries with average percentage. | Italy (1.90%), France (1R: 1.91%, 2R: 1.91%). |
| Countries with lower-than-average percentages. | Romania (1.28%) |
| Average value. | 2.85% |

Source: created by the authors using CLEA data. Note: Peruvian data are excluded due to unavailability.

*3.5. Comparison of Representativeness (Value of One Vote) in Overseas Electoral Districts*

The number of voters per parliament member can vary depending on the country and district. In this case, in constituencies where the number of voters per member is relatively small, the degree to which the voting behavior of a single voter can influence the election outcome (representativeness) is correspondingly high. In other words, the difference in the number of voters per parliament member indicates the influence of each vote that is cast by the voters. Conversely, in constituencies with many voters, the value of each vote is considered low.

As shown in Table 8, the number of voters per MP is relatively small in the overseas constituencies of Romania, Croatia, and Cape Verde, while the number of voters per MP is relatively large in the overseas constituencies of Italy and France. In other words, the representativeness (value of one vote) of overseas constituencies in Croatia and Cape Verde was higher than that in Italy and France.

**Table 8.** Groupings based on the representativeness (value of one vote) in overseas constituencies.

| Criterion | Countries Targeted |
| --- | --- |
| Countries with Above Average Representativeness. | Romania (29,272), Cape Verde (7447), Croatia (5287), Ecuador (47,626), Tunisia (40,089). |
| Countries with Average-Level Representativeness. | France (1R: 114,986, 2R: 115,022) |
| Countries with Below-Average Representativeness. | Italy (352,571) |
| Average. | 89,037 |

Source: created by the authors using CLEA data. Note: Peruvian data are excluded due to unavailability.

*3.6. Comparison of the Difference in Representativeness (Value of One Vote) between Domestic and Overseas Constituencies*

Finally, we compared the differences in representativeness (the value of one vote) between domestic and overseas constituencies in various countries. As shown in Table 9, the representativeness of domestic electoral districts in Italy, France, and Cape Verde surpassed that of overseas districts. However, in Croatia, Romania, and Ecuador, the opposite was true, with the representativeness in overseas electoral districts exceeding that of domestic districts. Additionally, in Tunisia, the difference in representativeness between domestic and overseas districts was 1.0, indicating almost no distinction in representativeness between the two.

**Table 9.** Groupings based on the difference in representativeness (value of one vote) between domestic and overseas constituencies.

| Criterion | Countries Targeted |
|---|---|
| Countries with higher domestic electoral constituencies. | Italy (0.213), France (1R: 0.175, 2R: 0.685), Cape Verde (0.616) |
| Countries with minimal domestic and overseas Differences. | Tunisia (1.002) |
| Countries with higher overseas constituencies. | Croatia (4.923), Romania (2.028), Ecuador (1.825) |
| Average. | 1.5 |

Source: created by the authors using CLEA data. Note: Peruvian data are excluded due to unavailability.

## 4. Discussion

The survey conducted in the eight countries included in this study compares the number of voters, voter turnout, and representativeness in both domestic and overseas electoral districts, revealing significant differences. The notable disparity in voter turnout in overseas electoral districts is particularly striking.

Croatia and Romania exhibit exceptionally high voter turnouts in their overseas electoral districts. In Croatia, while the domestic electoral district had a turnout of 72.94%, the overseas electoral district had an astonishing turnout of 99.92%. Similarly, in Romania, the domestic electoral district had a turnout of 39.44% compared to 94.46% in overseas districts. This difference likely reflects the high level of interest among overseas Croatian and Romanian voters in the voting process. This finding indicates that nationals living abroad in countries where voting is not mandatory, such as Croatia and Romania, actively engage in politics.

### 4.1. Relationship between Overseas Constituency Voter Turnout and Representativeness

A common characteristic that is shared by Croatia, Romania, and Ecuador, where the voter turnouts in overseas electoral districts are higher than average, is that in all three countries, the number of voters per representative in the overseas districts is lower than that in domestic districts. In other words, in the three countries with higher overseas voter turnouts, the representativeness in overseas districts surpasses that of domestic districts (refer to Table 10).

**Table 10.** Relationship between overseas constituency voter turnout and representativeness.

| Index | Countries | | | | | | | | Average |
|---|---|---|---|---|---|---|---|---|---|
| | Italy | Croatia | France (R1) | France (R2) | Romania | Cape Verde | Ecuador | Tunisia | |
| Voter turnout in overseas constituencies. | 29.84% | 99.92% | 19.11% | 16.44% | 94.46% | 36.69% | 46.60% | 15.52% | 44.82% |
| Domestic and overseas differences in voter turnout. | 43.11% | −53.90% | 30.33% | 27.04% | −55.02% | 33.60% | 35.19% | 25.97% | 10.79% |
| Domestic and overseas constituencies' ratio of representativeness (value of one vote) | 0.21 | 4.92 | 0.71 | 0.68 | 2.03 | 0.62 | 1.82 | 1.00 | 1.5 |

Source: created by the authors using CLEA data. Note 1: domestic and overseas voter turnout ratio = (voter turnout in domestic constituencies) − (voter turnout in overseas constituencies). If this value is negative, the turnout in overseas constituencies is higher than that in domestic constituencies. Note 2: domestic and overseas ratio of representativeness = (number of voters per member of parliament in domestic constituencies)/(number of voters per member of parliament in overseas constituencies). If this value is greater than 1, the representativeness (value of one vote) of overseas constituencies is greater than that of domestic constituencies. Note 3: Peruvian data are excluded, as they are unknown.

The difference in representativeness between domestic and overseas districts in each country can be evaluated through the ratio of the number of seats per 1 million voters by comparing domestic and overseas districts. The closer this ratio is to 1, the more equal the distribution of seats between domestic and overseas is, indicating that the value of a vote is nearly the same in both areas. Conversely, if the ratio of the number of domestic voters per representative (numerator) to the number of overseas voters per representative (denominator) is greater than one, overseas districts have fewer voters per representative, indicating a higher vote value than domestically. If this ratio is less than one, it indicates that the value of a vote is higher domestically.

On the other hand, in Italy, France, Tunisia, and Cabo Verde, where voter turnout in domestic districts is higher than that in overseas districts, the number of voters per representative in domestic districts is lower than in overseas districts, making the representativeness in domestic districts higher.

In summary, in Croatia, Romania, and Ecuador, where voter turnout in overseas districts is higher than average, the value of a single vote is greater in overseas districts than in domestic ones, making the impact of each voter's decision more significant in influencing election outcomes. In contrast, in Italy, France, Tunisia, and Cabo Verde, where voter turnout is higher in domestic districts, the value of a single vote is greater domestically, making each voter's decision more influential on domestic election outcomes.

Based on the surveyed data, this difference in representativeness (value of a single vote) between domestic and overseas districts may influence the voter turnout in overseas districts. That is, in overseas districts where the representativeness is higher than those in domestic ones, voters' decisions are more likely to influence election outcomes, possibly increasing their motivation to participate and consequently raising voter turnout.

*4.2. Relationship between Overseas Constituency Voter Turnout and Overseas Voter Ratio*

Additionally, in Croatia and Romania, where the turnouts in overseas constituencies are particularly high, the proportion of overseas voters in the total electorate is smaller than in the other five countries (see Table 11). This characteristic suggests that overseas voters in countries where they constitute a very small proportion of all voters have a higher incentive to actively participate in elections to reflect their minority opinions in national politics. Consequently, it can be argued that this may be linked to high voter turnouts.

**Table 11.** Relationship between overseas constituency voter turnout and overseas voter ratio.

| Index | Countries | | | | | | | | Average |
| --- | --- | --- | --- | --- | --- | --- | --- | --- | --- |
| | Italy | Croatia | France (R1) | France (R2) | Romania | Cape Verde | Ecuador | Tunisia | |
| Voter turnout in overseas constituencies. | 29.84% | 99.92% | 19.11% | 16.44% | 94.46% | 36.69% | 46.60% | 15.52% | 44.82% |
| Overseas voters among eligible voters. | 8.34% | 1.57% | 2.65% | 2.76% | 0.64% | 12.85% | 2.45% | 8.02% | 4.91% |

Source: created by the authors using CLEA data. Note: Peruvian data are excluded due to unavailability.

This point becomes even more persuasive when considered in conjunction with the representativeness of overseas constituencies (the value of one vote), as mentioned above. In Croatia and Romania, where the voter turnouts in overseas constituencies are particularly high, the number of voters per MP in overseas constituencies is lower than that in domestic constituencies, and the representativeness of the overseas constituencies is relatively high. Therefore, although overseas voters in Croatia and Romania are a minority in terms of the percentage of all voters, their voting behavior is likely to influence election results. Therefore, we believe that overseas voters in these two countries will have more incentives to actively participate in elections to have their minority opinions reflected in national politics, and that this will lead to higher voter turnouts in overseas constituencies.

## 5. Conclusions

This study examined the number of voters, quotas (number of seats), voting rates, and representativeness (number of seats) of eight countries that have introduced an overseas constituency representative system: France, Italy, Croatia, Romania, Ecuador, Peru, Cape Verde, and Tunisia. We conducted a comparative analysis based on data such as the value of votes and considered their characteristics.

By comparing data for domestic and overseas constituencies in the eight surveyed countries, it was found that the voter turnout in overseas constituencies varies greatly depending on the country. In particular, Croatia and Romania have shown very high turnout rates in their overseas constituencies. In Croatia, the turnout in domestic constituencies was 72.94%, whereas in overseas constituencies, it was extremely high at 99.92%. In Romania, the turnout in domestic constituencies was 39.44% compared with 94.46% in overseas constituencies. On the other hand, countries such as Italy, France, Tunisia, and Cape Verde tended to have significantly higher turnouts in domestic constituencies than in overseas constituencies. For instance, in Italy, the turnout was 72.94% in domestic constituencies compared with 29.84% in overseas constituencies.

In examining the factors influencing the differences in voter turnout rates in overseas constituencies, or between domestic and overseas voting rates, based on the results of this data analysis, we make two interesting observations. First, the difference in representativeness (the value of a single vote) between domestic and overseas constituencies can be identified as a factor that influences voter turnout in overseas constituencies. It can be hypothesized that in overseas constituencies where the representativeness is higher than in the domestic context, voters are more likely to influence the outcome, thereby potentially increasing their willingness to participate in elections and increasing voter turnouts. Second, overseas voters in countries where they constitute a very small proportion of all voters have a higher incentive to actively participate in elections to reflect their minority opinions in national politics, potentially leading to higher voter turnout rates than those within the home country.

Countries such as Croatia and Romania, where the voter turnouts in overseas constituencies are particularly high, have fewer voters per representative in overseas constituencies than in domestic ones, making the representativeness relatively higher in overseas constituencies. Therefore, despite being a minority in terms of their proportion in the total electorate, the voting behavior of overseas voters in Croatia and Romania is more likely to influence election outcomes. Consequently, overseas voters may have an even greater incentive to participate in elections to ensure that their minority opinions are reflected in national politics, which can explain the higher voter turnout rates in overseas constituencies in these countries.

This study contributes to the literature on overseas constituencies, which have rarely been analyzed in previous studies. The analysis of voter turnout in overseas constituencies in this study not only has implications for improving voter turnout in countries that have already introduced such systems but also offers suggestions for system design to achieve a high voter turnout in overseas constituencies in countries considering the introduction of similar systems in the future. Furthermore, the findings of this study may also have implications for the debate on the innovation of politics and th quality of democracy. The manifold effects of emigrant voting on home country politics can be expected to intensify due to increasing international migration and technological advances that facilitate emigrant linkages with their countries of origin. These effects range from swaying electoral results and government coalitions, influencing kin voters at home, and contributing to democratization processes and to the emergence of transnational party campaigning and infrastructure (Brand 2014; Gamlen 2015; Østergaard-Nielsen and Ciornei 2018). More importantly, the quality of democracy in the country of residence and origin may influence the turnout of emigrants in overseas constituencies in their homeland elections (Ciornei and Østergaard-Nielsen 2020).

Nevertheless, this study has some limitations that should be overcome by future research. This study focused on eight countries due to the limitation of the authors' research capacity, while there may be some more countries that used or have adopted a similar system, as a later study implies (Wellman et al. 2023). In addition, this study uses data on the latest election per country. We hope to obtain more robust results if a future study uses more elections in each country. In addition, these considerations are based on data analysis of representativeness and the number of overseas voters regarding the factors influencing the varying levels of voter turnouts in overseas constituencies. However, the determinants of voter turnouts may be more complex and context-dependent than the theories suggest (Stockemer 2017). The voter turnouts in overseas constituencies may have been influenced by other factors, as indicated by research on turnout and non-resident voters, such as the ease of election procedures, the eligibility of voters, access to the ballot, as well as non-resident citizens' economic, social, and cultural connections to their country of origin (Burgess and Tyburski 2020; Hutcheson and Arrighi 2015; Peltoniemi et al. 2023). The accuracy of the observations made in this study should be verified through detailed local and voter attitude surveys of overseas constituencies in Croatia, Romania, and other countries.

**Author Contributions:** S.Y.: Conceptualization (supporting); writing—original draft (lead); writing—review and editing (lead). T.S.: Conceptualization (lead); writing, review, and editing (supporting). All authors have read and agreed to the published version of the manuscript.

**Funding:** This study received no external funding.

**Institutional Review Board Statement:** Not applicable.

**Informed Consent Statement:** Not applicable.

**Data Availability Statement:** All data supporting the findings of this study have been included in this article.

**Conflicts of Interest:** The authors declare no conflict of interest.

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
