# Peer review of "Comparing the Election Systems for Overseas Constituency Representatives in Multiple Countries"

_socsci, doi:10.3390/socsci13030177_

Round 1

Reviewer 1 Report

Comments and Suggestions for Authors

The paper aims to provide a comparative analysis of overseas constituency representative systems in eight countries, focusing on specific key factors convincingly detalied on page 3. The more theoretical aim is to engage with the increasingly solid literature on turnout abroad. The paper represents an extremely solid research endeavor that can be published as such with only minor comments. Specifically, it is important to explain the focus on these countries more explicitly, as references such as "including France and Italy" in the abstract may not be easily understood. I suggest providing a clearer rationale for this selection. 

I appreciated the quality of the data and the rigorous analysis presented. However, I feel that the contextual background is almost entirely absent. Beyond the institutional dimension, these eight countries require some context to fully comprehend the solid discussion provided, particularly regarding potential interactions with previous studies on the variable of reference, namely turnout, in analyses dealing with these eight cases. This is the main relevant comment I have for this otherwise very solid paper, which, in my view, can become a significant contribution to the literature.

A minor comment pertains to the need to add a couple of theoretical references in the conclusions to broaden the existing discussion. As it stands, the paper remains exclusively limited to empirical assessment, but I believe its solidity can be reinforced by introducing some more ambitious theoretical aspects in the last paragraph, along with interaction with previous studies on this topic.

Author Response

Thank you very much for taking the time to review this manuscript. Please see the attached file for my response.

Reviewer 2 Report

Comments and Suggestions for Authors

I would like to thank the authors for submitting this interesting paper. The paper’s central finding --- the difference in representativeness between domestic and diaspora constituencies matters for overseas voters’ electoral participation --- is illuminating. I imagine that many scholars would build on this paper and use this concept as both independent and outcome variables.

Now the literature on diaspora turnout is burgeoning. This includes cross-national studies by Burgess and Tybuski (2020) and Ciornei and Eva Østergaard-Nielsen (2020). There are also a number of country- or region-specific case studies. I think that the current version of ms does not sufficiently engage with this literature. The paper should provide a proper literature review of the existing studies related to diaspora voting.

Further, Hutcheson and Arrighi (2015) provide a useful framework to classify different aspects of diaspora electoral systems. Although the paper talks about different voting rules, it does not discuss other important dimensions in which different countries’ diaspora electoral systems vary (e.g., eligibility criteria, ballot access, # polling stations, etc.). I think it is better to engage with Hutcheson and Arrighi (2015). This is important because it would allow the paper to assess if the difference in representativeness is correlated with some aspects of diaspora electoral systems that are currently ignored in the ms.

The paper should elaborate on its case selection criteria. According to Wellman et al. (2023), around twenty countries have diaspora legislative seats. Why does the paper focus only on eight cases? Also, the paper uses one election per country. Why not go back further and use more elections in each country?

I was puzzled by the extremely high turnout rates in Croatia and Romania and dug into the case of Croatia by myself. According to the information on Wikipedia (https://en.wikipedia.org/wiki/Electoral_district_XI_(Croatian_Parliament)), it seems to be the case that there was a huge change in turnout between 2011 and 2015, and this is explained by a large drop in the denominator --- # registered voters went down from 411,758 in 2011 to 28,944 in 2015 and further to 21,223 in 2016. This point raises two questions. First, in the case of Croatia, how did this large change in eligible diaspora voters happen? Did it change the registration requirements before/after 2015? Second, is the meaning of the denominator the same across the countries analyzed in the paper? I am kind of worried that in some countries, the denominator includes all voting-age persons living abroad, while in other cases, the denominator includes only overseas voters who registered to vote. If this is the case, turnout rates across different countries mean different things, which makes cross-national comparison difficult.

References

Burgess, Katrina, and Michael D Tyburski. 2020. "When parties go abroad: Explaining patterns of extraterritorial voting." Electoral Studies 66:102169.

Ciornei, Irina, and Eva Østergaard-Nielsen. 2020. "Transnational turnout: Determinants of emigrant voting in home country elections." Political Geography 78:102145.

Hutcheson, Derek S., and Jean-Thomas Arrighi. 2015. “Keeping Pandora's (ballot) box half-shut”: A comparative inquiry into the institutional limits of external voting in EU Member States." Democratization 22(5):884-905.

Wellman, Elizabeth Iams, Nathan W Allen, and Benjamin Nyblade. 2023 "The extraterritorial voting rights and restrictions dataset (1950–2020)." Comparative Political Studies 56(6):897-929.

Author Response

(The authors gave the same response as above.)

Round 2

Reviewer 2 Report

Comments and Suggestions for Authors

I appreciate the authors' responses to my comments. The revisions did not necessarily go as far as I would have liked. For example, I was expecting (i) a more extended review of prior research on diaspora electoral participation; (ii) extending the analysis to more than one election per country; (iii) empirically examining how representativeness is correlated with other factors impacting diaspora turnout that prior work identifies (e.g., access to polling stations); and so on. But I suppose this is what the journal expects. Also, given that the paper focuses more on theory building than testing, I guess I was expecting too much.

Author Response

Thank you very much for taking the time to review revised manuscript. Please see the attached file for my response.
